# The Evolution of Immune Checkpoint Inhibitors in Advanced Urothelial Carcinoma

**DOI:** 10.3390/cancers14071640

**Published:** 2022-03-24

**Authors:** Hélène Houssiau, Emmanuel Seront

**Affiliations:** Department of Medical Oncology, Centre Hospitalier de Jolimont, Rue Ferrer 159, 7100 Haine Saint Paul, Belgium; helene.houssiau@student.uclouvain.be

**Keywords:** urothelial carcinoma, immune checkpoint inhibitors, maintenance strategy, PD-L1, tumor mutation burden, combined positive score

## Abstract

**Simple Summary:**

Urothelial carcinoma is an aggressive cancer with a high risk of metastatic progression. Chemotherapy plays a key role in the management of metastatic urothelial carcinoma, with, however, no possibility of cure. Immune checkpoint inhibitors have significantly improved the outcomes of patients, delaying progression of disease and improving quality of life. However, many questions remain concerning the optimal use of immunotherapy in urothelial carcinoma: When to start? Which biomarker of sensitivity/resistance to use? Which of the available options will increase the efficacy of immune checkpoint inhibitors? We review the mechanisms of immune checkpoint inhibitors as well as the current management of patients with metastatic urothelial carcinoma in the era of immunotherapy.

**Abstract:**

Urothelial carcinoma is an aggressive cancer and development of metastases remains a challenge for clinicians. Immune checkpoint inhibitors (ICIs) are significantly improving the outcomes of patients with metastatic urothelial cancer (mUC). These agents were first used in monotherapy after failure of platinum-based chemotherapy, but different strategies explored the optimal use of ICIs in a first-line metastatic setting. The “maintenance” strategy consists of the introduction of ICIs in patients who experienced benefit from first-line chemotherapy in a metastatic setting. This allows an earlier use of ICIs, without waiting for disease progression. We review the optimal management of mUC in the era of ICIs, based on the key clinical messages arising from the pivotal trials.

## 1. Introduction

The immune system is able to detect and inhibit cancer cell progression. Tumor cells release tumor-associated antigens, named neoantigens, that are captured by antigen-presenting cells (APCs) through the major histocompatibility complex (MHC) I. APCs migrate to lymph nodes, where they activate effector T-cells by presenting these neoantigens to the surface T-cell receptors (TCRs). In this priming phase, activated T-cells are able at this time to recognize any cancer cell synthetizing these neoantigens and to infiltrate tumors with subsequent inflammatory reactions. The binding of TCRs with MHCs can regulate the immune system through equilibrium in positive (co-stimulatory signals) and negative (co-inhibitory signals) interactions that arise between the T-cells and the APCs. The main positive regulatory interaction is provided by the binding of B7 on APCs to CD28 on T-cells, leading to the recruitment and activation of Src-like tyrosine kinases, which, in turn, activates the phospho-inositol-3 kinase (PI3K)–Akt (protein kinase B)–mammalian target of rapamycin (mTOR) cascade, the mitogen-activated pathway kinase (MAPK)–ERK axis and the Janus kinase (JNK) signaling pathway. One of the main resultant effects is the production of interleukin-2 (IL-2) through NF-kβ activation. IL-2 promotes immune response by stimulating CD4+ T-cell proliferation and differentiation into helper T-cells (Th), including Th1 and Th2 cells, and increasing the number and activity of CD8+ T-cell effector and natural killer cells (NKs). During the effector phase, activated T-cells are trafficked to specific sites by following a chemokine gradient; on contact with tumor cells, they release interferon gamma (IFN-γ) and other cytokines, promoting cytotoxicity [1,2].

However, tumor cells develop progressive adaptative mechanisms that allow inactivation of T-cell activity and impaired immune defenses. One of these adaptative mechanisms involves immune checkpoint proteins that are located on cell surfaces and that physiologically disrupt immune reactions. Many checkpoint proteins exist, including the cytotoxic T-lymphocyte-associated protein 4 (CTLA-4) located on APC surfaces, the programmed cell death protein-1 (PD-1) located on T-cells and the programmed death ligand-1 (PD-L1) located on tumor cells. CTLA-4 exerts its inhibitory effect by competing with CD28 and binding to B7, resulting in T-cell inactivation in lymphoid tissues (Figure 1). Among mechanisms leading to immune checkpoint overexpression, excessive activation of oncogenic pathways (the PI3K–Akt axis or the MAPK cascade) in tumor cells can induce upregulation of PD-L1 on their surfaces. In turn, PD-1 on T-cell surfaces can relieve inhibition of PTEN, a phosphatase that physiologically prevents the activation of Akt by PI3K. PD-1 can also negatively regulate the expression of other cell surface receptors, such as TCRs, which results in impaired recognition of MHC by T-cells. Furthermore, PD-1 signaling can induce blockade of T-cell cycle progression by enhancing activation of the cyclin-dependent kinase (Cdk) inhibitor p27kip1. Activation of PD-1 during the stimulation of T-cells via TCRs and CD28 can also inhibit the downstream activation of the PI3K–Akt and MAPK pathways, blocking the enzymatic machinery responsible for cell cycle progression (Figure 2) [3,4,5].

Immune checkpoint inhibitors (ICIs) are monoclonal antibodies that, by targeting immune checkpoints and preventing their binding to their ligands, disrupt the inactivation of T-cells. The most studied ICIs are two anti-CTLA-4s (ipilimumab and tremelimumab), two anti-PD-1s (nivolumab and pembrolizumab) and three anti-PD-L1s (atezolizumab, durvalumab and avelumab). Avelumab can, in addition to its PD-L1 inhibition on tumor cells through the Fab domain, bind to effector cells with the Fc region, leading to an active lysis of the tumor cells mediated by these effector cells. This antibody-dependent cell-mediated cytotoxicity (ADCC) is not observed with other ICIs. New antibodies are currently developed to target other immune checkpoints, such as T-cell immunoglobulin and mucin-domain containing-3 (TIM-3), lymphocyte activation gene-3 (LAG-3) and T-cell immunoglobulin and ITIM domain (TIGIT) [6,7,8].

## 2. ICIs in Urothelial Carcinoma: Rationale for Clinical Efficacy

Urothelial carcinoma (UC) appears to be a good candidate for ICIs. First, as already described, neoantigen release represents an important step in the immunogenicity of tumors. The generation of neoantigens that are recognized as ‘non-self’ by immune cells (ICs) could enhance anti-tumor immune response. Production of neoantigens is closely related to mutations occurring during tumor cell proliferation and UC carries one of the highest mutation rates among cancer types [9,10]. Second, lymphocyte infiltration may be associated with better outcomes in non-muscle invasive UC (NMIUC) and explain why intravesical instillations of bacillus Calmette–Guerin (BCG), by enhancing lymphocyte infiltration and inducing local inflammation, were shown to prevent recurrences [11]. It was also shown that high levels of tumor-infiltrating CD8+ lymphocytes (TILs) in muscle invasive UC were associated with better disease-free survival and overall survival (OS) than low levels of TILs [12]. Third, some studies showed that levels of PD-L1 expression in vesical UC were correlated with higher-stage, higher frequencies of postoperative recurrence and poorer survival [13]. Based on these observations, ICIs appear as promising agents in UC, which remains a challenge for clinicians. UC is the most frequent histology and accounts for more than 90% of bladder cancer. Even if 75% of patients present with superficial and NMIUC with an excellent outcome after local treatment, invasion of muscle (MIUC) is associated with worse prognosis and a high rate of recurrence despite multimodal strategies, including cystectomy and perioperative chemotherapy [14,15,16]. Patients with metastatic UC (mUC) have poor outcomes and, based on historical treatment, an OS that does not exceed 15 months. For many years, cisplatin-based chemotherapy remained the standard treatment in a first-line metastatic setting and second-line options were limited, including single agents, such as paclitaxel, vinflunine and docetaxel [17,18,19].

## 3. ICI after Failure of Platinum-Based Therapy: Better Than Chemotherapy

ICIs have been evaluated in late-stage UC disease, after failure of platinum-based therapy; since 2016, five ICIs have been approved in this setting (Table 1).

Atezolizumab (1200 mg administered intravenously (IV) every 3 weeks) received approval based on the results of the phase II IMvigor210 trial Cohort 2. This non-randomized trial enrolled 310 mUC patients who were heavily pretreated (20% received ≥3 previous chemotherapy regimens). These patients were stratified according to tumor-infiltrating immune cell (IC) PD-L1 = IC0 (<1%), IC1 (≥1% but <5%) and IC2/3 (≥5%). Atezolizumab resulted in an ORR of 16%, including 7% complete response (CR) in all patients. PD-L1 expression was associated with higher response (ORR of 28%, including 15% CR in IC2/3 patients). Interestingly, 10% of responses were also seen in PD-L1-negative patients. The median OS reached was 7.9 months for all patients and 11.9 months for IC2/3 patients [20]. These first results were impressive in regard to the number of previous treatments these patients received and given that treatment options were limited in UC, reflecting the promising role of ICIs in advanced UC.

The randomized phase III IMvigor211 trial compared atezolizumab to standard second-line chemotherapy (vinflunine, paclitaxel, docetaxel) in 931 mUC patients after failure of platinum-based chemotherapy. The primary efficacy endpoint OS was tested in a hierarchical approach and statistical significance needed to be achieved in the IC2/3 population in order to evaluate statistically the further subgroups, such as the ITT population. Atezolizumab failed to demonstrate improved median OS compared to chemotherapy in IC2/3 PD-L1-expression patients (11.1 vs. 10.6 months; hazard ratio (HR) 0.87; *p* = 0.41); there was no difference in the 1-year OS rate (46% vs. 41%, respectively). A moderate but significant difference in OS was observed in the ITT population treated with atezolizumab compared to chemotherapy (8.6 vs. 8.0 months; HR 0.85; *p =* 0.038). These disappointing results were mainly explained by the fact that the OS in the chemotherapy arm, and particularly in the vinflunine arm, was higher than expected. However, despite the unmet primary endpoint, this trial confirmed the longer median duration of response (DOR) with atezolizumab compared to chemotherapy in the overall population (21.7 vs. 7.4 months, respectively), confirming the possibility of a long-lasting response in responding patients [21]. Despite the absence of superiority of atezolizumab compared to chemotherapy and the absence of level 1 evidence, these results suggest that atezolizumab could be proposed after failure of platinum-based chemotherapy in mUC.

Nivolumab was approved based on a non-randomized phase II trial, the Checkmate 275, which enrolled 265 previously treated patients with mUC. Nivolumab (3 mg/kg IV every 2 weeks) resulted in 20% ORR for the total population. PD-L1 expression was associated with higher ORR (28.4% for patients with high (≥5%), 23.8% for patients with ≥1% and 16% for patients with low (<1%) tumor cell (TC) PD-L1 expression. The median OS was 8.74 months in all patients and was higher in patients with PD-L1 ≥1% compared to patients with <1% (11.3 vs. 5.95 months, respectively) [22].

Pembrolizumab is the only agent that was shown to improve survival in mUC after platinum-based chemotherapy, based on the KEYNOTE-045 trial, a randomized phase III trial that compared the efficacy of pembrolizumab (200 mg IV every 3 weeks for up to 2 years) to chemotherapy (docetaxel, paclitaxel or vinflunine) in 524 mUC patients. Pembrolizumab improved OS in all patients compared to chemotherapy (10.1 vs. 7.2 months, respectively; HR 0.71, 95% confidence interval (CI) 0.59–0.86). There was no significant difference in PFS between pembrolizumab and chemotherapy in all patients (2.1 vs. 3.3 months, respectively; HR 0.98; 95% CI 0.81–1.19; *p* = 0.42). Pembrolizumab also increased ORR compared to chemotherapy (21.1% versus 11.4%), with, after a follow-up of more than 5 years, a longer median DOR (29.7 vs. 4.4 months, respectively). The combined positive score (CPS) represents the percentage of PD-L1 in ICs and TCs related to the numbers of TCs; positive CPS (≥10) was not associated with a better OS, PFS or ORR in the pembrolizumab arm. OS benefit with pembrolizumab was observed in all subgroups of patients, regardless of age, ECOG performance status, prior therapy and chemotherapy choice [23,24]. As usually observed with ICIs, pembrolizumab was better tolerated than chemotherapy, with grade 3–4 toxicities not exceeding 15% (vs. 49.4% with chemotherapy) [23]. Pembrolizumab thus represents level 1 evidence in a second-line setting, after failure of platinum-based therapy in mUC, regardless of CPS.

Two other ICIs, durvalumab and avelumab, also received FDA approval based on Phase I/II trials. Durvalumab (10 mg/kg IV every 2 weeks) resulted in 191 mUC patients progressing to platinum-based treatment, with an ORR of 18% (3% CR); furthermore, the median OS was 18.2 months in the entire population and was higher in high PD-L1 compared to low or negative patients (20 vs. 8.1 months). Avelumab (10 mg/kg IV every 2 weeks) resulted in similar results with an ORR of 16.5%, including 4.1% of CR and 12.4% of partial response (PR). Avelumab showed clinical benefit in high-risk subgroups, including the elderly, those with renal insufficiency and upper tract patients [25,26,27,37] (Table 1).

Based on all these trials, ICIs play a key role after platinum-based therapy and could be considered standard of care in this setting.

## 4. Determining the Optimal Place of ICIs: Moving towards the First-Line Setting

### 4.1. ICI in Maintenance Represents the New Standard Strategy

Historically, platinum-based chemotherapy has remained the standard of care for many years, followed, at progression, by ICI monotherapy. The maintenance strategy, meaning starting ICI directly after first-line chemotherapy without waiting for disease progression, is now the current standard of care in 2022, approved by the FDA and European Medical Agency (EMA). This approval was based on the results of the randomized phase III JAVELIN Bladder 100 trial. This study enrolled 700 patients with unresectable locally advanced or mUC who had not experienced disease progression after a first-line platinum-based chemotherapy. Enrolled patients had to present CR, PR or stable disease (SD) after four to six cycles of cisplatin or carboplatin plus gemcitabine. Patients were randomized (1:1) to receive maintenance treatment with avelumab (10 mg/kg IV every 2 weeks) and best supportive care (BSC) or only BSC. The primary endpoint was OS, which was assessed in both the overall population and the PD-L1-positive population. Fifty-one patients of patients were PD-L1-positive. (PD-L1 was considered to be positive if at least one of the following three criteria were met: 1) at least 25% of TCs positive, 2) at least 25% of ICs positive if more than 1% of the tumor area contained ICs or 3) 100% of ICs positive if no more than 1% of the tumor area contained ICs.) OS was significantly improved with avelumab + BSC compared to BSC alone in the overall population (21.4 vs. 14.3 months; HR, 0.69; 95% CI 0.56–0.86; *p* = 0.0005). The 12-month OS was 71.3% in the avelumab maintenance group compared to 58.5% in the BSC group. In PD-L1-positive patients, OS was also significantly improved in the avelumab maintenance group compared to the BSC group (not reached (NR) vs. 17.1 months; HR 0.56; 95% CI 0.40–0.79; *p* = 0.0003), as well as the 12-month OS (79.1% vs. 60.4%, respectively). The PFS was increased in the overall population, with the maintenance avelumab strategy compared to the control group (3.7 vs. 2 months; HR 0.62; 95% CI 0.52–0.75), and in the PD-L1-positive patients (5.7 vs. 2.1 months; HR 0.56; 95% CI 0.43–0.73). The fact that 43.7% of patients randomized to the BSC arm were further treated with ICIs highlights the importance of introducing avelumab before progression of the disease, directly after chemotherapy [28].

Subgroup analyses showed OS benefit irrespective of duration/cycles of chemotherapy and the amplitude of response obtained with chemotherapy (CR/PR or SD). This suggests that patients may receive four or/to six cycles with a similar benefit from avelumab. In clinical practice, the number of cycles has thus to be decided case by case, based on tolerance and response induced by chemotherapy. It may be that a patient who does not tolerate chemotherapy and who gets only a SD after four cycles should directly go for avelumab in maintenance. Conversely, a patient who gets a PR/CR after four cycles should complete the six cycles of chemotherapy before receiving avelumab. Avelumab maintenance also improved the survival of patients, regardless of the response obtained with chemotherapy; avelumab maintenance resulted in HRs of 0.81 in CR patients and 0.62 in patients with PR, suggesting that even in the absence of residual radiological disease after chemotherapy, avelumab should be proposed. The benefit was also observed regardless of the location of tumors (upper or lower tract tumors) or the initial stage at diagnosis. Unanswered questions remain, including the optimal number of cycles of avelumab in maintenance strategies; in the JAVELIN Bladder 100 trial, the median duration of avelumab reached 24.9 weeks (range from 2.0 to 159.9 weeks). Real-world observations will help to answer this question. Importantly, the tolerability of avelumab was supported by patient-reported outcome (PRO) data, and there was no deterioration in quality of life [38,39]. The AE profile was similar to that observed in ICI trials in mUC. AEs of any grade occurred in 98.0% in the avelumab group and in 77.7% in the control group, with adverse events of grade ≥3 occurring in 47.4% and 25.2%, respectively. Discontinuation of treatment induced by AE occurred in 11.9% of patients in the avelumab group [28] (Table 1). Figure 3 summarizes the current algorithm in the management of first-line metastatic UC patients. The fact that carboplatin does not alter the subsequent efficacy of avelumab maintenance and that four cycles of chemotherapy were found to be similar to six cycles is of benefit for patients who are deemed ineligible for cisplatin and/or who tolerate chemotherapy poorly. Early introduction of avelumab in maintenance, without waiting for disease progression, is particularly interesting in patients with a high tumor burden who experienced only stable disease with chemotherapy.

### 4.2. Combination of Chemotherapy and ICI Does Not Improve Survival in a First-Line Setting

The IMvigor130 trial is a multicentric phase III trial that randomized patients with locally advanced or mUC in three groups: atezolizumab (1200 mg IV every 3 weeks) plus chemotherapy (cisplatin/carboplatin plus gemcitabine—group A); atezolizumab (1200 mg IV every 3 weeks alone—group B) and placebo plus a chemotherapy regimen (group C). Atezolizumab was pursued until radiological progression or unacceptable toxicity. The co-primary endpoints were PFS and OS between groups A and C and OS between groups B and C in a hierarchical approach; this last endpoint was statistically evaluated only if the OS of group A was statistically superior to group C.

In the ITT population, PFS tended to be increased in group A compared to group C (8.2 vs. 6.3 months, respectively; *p* = 0.007). There was no statistical difference in median OS between groups A and C, with a median survival of 16 months and 13.4 months, respectively (HR 0.83; 95% CI 0.69–1.00; *p* = 0.027, but not crossing the interim efficacy boundary of 0.007). The combination of chemotherapy plus ICI did not result in a higher ORR compared to chemotherapy alone (47% vs. 44%, respectively). Cisplatin chemotherapy, high expression of PD-L1 and high ECOG performance status were associated with greater benefit [29].

In a similar design, the phase III trial KEYNOTE-361 randomized 1000 patients with untreated locally advanced and unresectable or mUC in three arms: pembrolizumab plus chemotherapy (six cycles of cisplatin/carboplatin plus gemcitabine—group A), pembrolizumab monotherapy (200 mg IV every 3 weeks—group B) and chemotherapy (six cycles of cisplatin/carboplatin plus gemcitabine—group C). Pembrolizumab was given for up to 35 cycles. As observed in the IMvigor130 trial, the addition of pembrolizumab to first-line platinum-based chemotherapy did not significantly improve PFS or OS in the total population per the prespecified *p*-value boundaries of 0.0019 and 0.014, respectively. The response rate reached 54.7% in group A and 44.9% in group C, including a CR rate of 15% and 12%, respectively. The median DOR was 8.5 months (8.2–11.4) in group A and 6.2 months (5.8–6.5) in group C. Due to the lack of significance, no further formal statistical hypothesis testing was performed [30]. The combination of chemotherapy and ICI did not result in an increased rate of toxicity. In the IMvigor130 trial, grade 3 and 4 AEs were observed in 81% of patients treated with the combo and in 81% of patients treated with chemotherapy alone. In KEYNOTE-361, grade 3 and 4 AEs were observed in 87% in the combo group and in 82% of patients in the chemotherapy group [30].

These two phase III randomized trials do not support the combination of ICI + chemotherapy in a first-line metastatic setting. ICI was given in maintenance in these two trials but did not result in OS benefit, as observed in the JAVELIN Bladder 100 trial [28]. Multiple reasons could explain this discrepancy. First, the JAVELIN 100 trial enrolled only patients with CR/PR/SD on chemotherapy, selecting patients with better prognosis. Second, concomitant administration of chemotherapy could induce potential immunosuppressive effects that could decrease the efficacy of ICI. Third, cisplatin was more frequently used in the JAVELIN bladder 100 trial compared to the IMvigor130 and KEYNOTE-361 trials, resulting in greater antitumoral effects (Table 1).

Based on these results, there is no evidence to support the use of chemotherapy plus ICI in first-line metastatic disease.

### 4.3. Superiority of ICI over Platinum-Based Chemotherapy Has Not Been Demonstrated in A First-Line Metastatic Setting

Three randomized trials evaluated the role of ICI monotherapy in a first-line setting. The IMvigor130 and KEYNOTE-361 trials compared atezolizumab and pembrolizumab alone, respectively, to chemotherapy. However, these results were purely exploratory, as in both trials, following the hierarchical approach, the combination of ICI plus chemotherapy appeared not statistically superior to chemotherapy in terms of PFS and OS.

In the IMvigor130 trial, atezolizumab resulted in a superior median OS compared with placebo plus chemotherapy (15.7 vs. 13.1 months; HR 1.02; 95% CI 0.83–1.24), but during the first months of treatment, patients were more likely to survive in the chemotherapy group, due to higher response rates observed with chemotherapy compared to atezolizumab (44% vs. 23.4%, respectively). However, for responding patients, the DOR was longer for atezolizumab monotherapy than for chemotherapy (29.6 vs. 8.1 months, respectively). Even if no formal statistical comparison could be done, exploratory subgroups analyses demonstrated that the median OS for the PD-L1 IC2/3 (defined as PD-L1 expression on ICs ≥5% of IC) patients was higher in the atezolizumab monotherapy arm than in the chemotherapy arm (27.5 vs. 16.7 months, respectively) [29].

The KEYNOTE-361 trial had a similar design to the IMvigor130 trial. In the ITT population, there was no important difference in the median OS between pembrolizumab and chemotherapy (15.6 months vs. 14.3 months, respectively). As observed in the IMvigor130 study, chemotherapy resulted in a higher ORR than pembrolizumab (30.3% vs. 44.9%, respectively), explaining the superiority of chemotherapy in the first months of treatments. Even if exploratory, high CPS (≥10) was not associated with greater OS benefit with pembrolizumab compared to chemotherapy (16.1 vs. 15.2 months, respectively) [30].

The phase III trial DANUBE randomized mUC patients in three arms: durvalumab (1500 mg IV every 4 weeks) monotherapy, durvalumab plus tremelimumab and chemotherapy. The co-primary endpoints were OS compared between durvalumab monotherapy and chemotherapy alone in PD-L-positive patients (≥25% of TCs or ICs). There was no significant difference in the median OS between durvalumab and chemotherapy in the PD-L1-positive population (14.4 vs. 12.1 months, respectively; HR: 0.89; 95% CI 0.71–1.11; *p* = 0.30). Similar to atezolizumab and pembrolizumab monotherapy, durvalumab efficacy appeared late compared to chemotherapy, due to a lower response rate with durvalumab (26% vs. 46%, respectively). The median PFS in the ITT population was 2.3 months in the durvalumab group compared to 6.7 months in the chemotherapy group. In the PD-L1-positive population, median PFS was 2.4 months and 5.8 months, respectively [31] (Table 1).

These three trials did not show any superiority for ICI monotherapy compared to platinum-based chemotherapy in patients with mUC in a first-line setting. ICI should not be considered in patients deemed eligible for platinum-based chemotherapy, particularly in case of high tumor burden if rapid important response is needed (Figure 3). Once again, the results of the JAVELIN Bladder 100 trial favor the introduction of platinum-based chemotherapy (cisplatin or carboplatin) in a first-line setting followed by avelumab maintenance.

### 4.4. Strategies in Cisplatin-Ineligible Patients: Monotherapy or Maintenance ICI

A proportion of patients are deemed ineligible for cisplatin; criteria for cisplatin ineligibility include at least one of the following items: creatinine clearance >30 and <60 mL/min, ≥G2 hearing loss or peripheral neuropathy or ECOG performance status (PS) 2. The antitumoral efficacy of carboplatin appears lower than that of cisplatin, with an ORR ranging between 30 and 41%, PFS reaching only 5.8 months and OS not exceeding 9.3 months [40].

Two phase II trials were dedicated to evaluating ICI monotherapy in cisplatin-ineligible patients who had received no prior chemotherapy in a metastatic setting. Cohort 1 of the phase II IMvigor210 study enrolled 119 patients; 70% had renal impairment, 56% and 15% had 1 and 2 Bajorin risk factors (poor performance status and visceral metastases), respectively, and 32% of patients had high PD-L1 expression (IC2/3). After a median follow-up of 70.8 months, the ORR was 23.5% in all patients, including 8% CR. The median DOR had not been reached in all patients or in pre-defined PD-L1 subgroups (range 3.7 to 21.0). At the time of the current analysis, 54% of the responders in the total cohort had an ongoing response (47.4% in the PD-L1-low and 66.7% in the PD-L1-high subgroups). The median PFS was 2.7 months (95% CI 2.1–4.2) in all patients, 4.1 months (95% CI 2.3–11.8) in IC2/3 patients, 2.1 months (95% CI 2.1–5.4) in IC1 patients and 2.6 months (95% CI 2.1–5.7) in IC0 patients. The median OS was 16.3 months (95% CI 10.4 to not estimable) in all patients, 12.3 months (95% CI 6.0 to not estimable) in IC2/3 patients and 19.1 months (95% CI 9.8 to not estimable) in IC0/1 patients. The 12- and 60-month survival rates were 57% and 21.6% in all patients, respectively [20,32,33].

The phase II trial KEYNOYTE-052 evaluated pembrolizumab as a first-line agent in 370 cisplatin-ineligible patients (28.9% were ≥80 years, 41.9% of patients with ECOG PS 2, 49% with renal dysfunction, 35% with ECOG PS 2 and visceral metastatic disease and 10% with ECOG PS 2 and renal dysfunction). These patients received pembrolizumab 200 mg IV every 3 weeks for up to 24 months. The primary endpoint ORR reached 28.6%, including 8.9% CR and 19.7% PR. The disease control rate, combining CR, PR and SD reached 46.8%. After a median time from enrollment to data cut-off of 11.4 months (range, 0.1–41.2 months), the median DOR was 30.1 months (95% CI 18.1 months NR); responses lasted ≥12 and ≥24 months in 67% and 52% of patients, respectively. The median OS was 11.3 months (95% CI, 9.7–13.1), with 12- and 24-month OS rates of 46.9% and 31.2%, respectively. CPS positivity was associated with ORR and OS; in PD-L1 CPS ≥10 patients, ORR was 47.3% (CR 20.0% and PR 27.3%) and median OS was 18.5 months (95% CI, 12.2–28.5 months). In PD-L1 CPS < 10, ORR was 20.3% and median OS was 9.7 months (95% CI, 7.6–11.5 months). The 24-month OS rates were 47.0% and 24.0% for high CPS and low CPS patients, respectively [34,35] (Table 1).

Head-to-head comparison with a carboplatin-based regimen is not feasible; however, the higher response rate observed with carboplatin suggests the preferability of a carboplatin regimen over ICI in cisplatin-ineligible patients with high tumor burdens. This strategy is also supported by the JAVELIN Bladder 100 results. Benefit was improved both with cisplatin and with carboplatin given avelumab maintenance; the HR for patients receiving avelumab maintenance compared to BSC after cisplatin- and carboplatin-based regimens was 0.69 (0.51–0.94) and 0.66 (0.47–0.91), respectively [28,38,39].

A minority of mUC patients are deemed ineligible for carboplatin in a frontline setting; carboplatin ineligibility characterizes patients with poor general status (ECOG PS >3), creatinine clearance <30 mL/min), grade >3 peripheral neuropathy and severe heart failure. In these patients, as options are very limited, ICI could be proposed in a first-line setting, with higher benefit expected in PD-L1-positive patients (Figure 3).

### 4.5. No Current Place for ICI–ICI Combinations in mUC

ICI combinations, particularly the combination of PD-L1–PD-1 inhibitors and CTLA-4 inhibitors, improve the outcomes of some cancers, such as melanoma, mesothelioma and renal cancer. CTLA-4 is expressed by regulatory memory CD4+ and T-cells and is functional during the priming phase. The PD-1–PD-L1 interaction occurs mainly in the effector phase. The combination of CTLA-4 and PD-L1 inhibitors has been evaluated in the randomized phase III DANUBE trial. This trial compared three regimens: a tremelimumab–durvalumab combination for up to four doses followed by durvalumab in maintenance, durvalumab monotherapy (1500 mg IV every 4 weeks) and six cycles of a platinum-based chemotherapy regimen (gemcitabine combined to carboplatin/cisplatin). The tremelimumab–durvalumab combo did not significantly improve the outcomes of patients; in the ITT population, the median OS was not significantly different from chemotherapy (15.1 vs. 12.1 months, respectively; HR 0.85; *p* = 0.075); ORR was even lower with this combination compared to chemotherapy (36 vs. 46%, respectively). Even if the primary endpoint was not met, these outcomes were analyzed in PD-L1-high patients (≥25% of PD-L1-positive TCs or ICs) in an exploratory way. Interestingly, tremelimumab–durvalumab resulted in a higher OS in these patients (17.9 vs. 12.1 months, respectively; HR 0.74) and a similar ORR (48% vs. 47%, respectively) compared to chemotherapy, suggesting that this ICI–ICI combo should be further explored in the light of more promising biomarkers [31]. However, in current clinical practice, there is no place for ICI combinations in a first-line metastatic setting in patients who are cisplatin-eligible.

Other regimens and dosage schedules have been tested in early clinical trials. The CheckMate 032 trial randomized mUC previously treated patients in three groups: nivolumab 1 mg/kg + ipilimumab 3 mg/kg (N1/I3) or nivolumab 3 mg/kg + ipilimumab 1 mg/kg (N3/I1) every 3 weeks for four cycles, followed by nivolumab 3 mg/kg every 2 weeks or nivolumab monotherapy 3 mg/kg (N3) every 2 weeks. N1/I3 induced a higher response rate compared to the other cohorts (38.5% vs. 26% for N3/I1 and 25.6% for N3). The median PFS in the N1/I3 group and in the N3/I1 group was 4.3 months and 2.6 months, respectively, and the median OSs were 10.2 months and 7.3 months, respectively. A longer follow-up is of course required. The rates of grade 3–4 AEs were similar in each group, at 30.8% and 31.7% for the N1/I3 and N3/N1 arms, respectively [36].

In a single-arm phase II trial (NCT01524991), 36 patients received two cycles of cisplatin–gemcitabine alone followed by four cycles of gemcitabine, cisplatin and ipilimumab. The ORR reached 69% with a median OS of 14.6 months, which was not superior to historical results for cisplatin–gemcitabine alone. Interestingly, translational analysis showed that the addition of ipilimumab increased the proportion of CD4+ and CD8+ T-cells without depleting T-regulatory cells, highlighting the rationale for testing these combinations in larger cohorts [41].

## 5. Optimizing Biomarker Profiles in UC

All these trials showed that only a proportion of patients could benefit from ICIs and that PD-L1 expression is not able to predict efficacy or inefficacy of ICIs in metastatic UC. Even if a high amplitude of benefit may be expected in PD-L1-positive patients, response can also be observed in PD-L1-negative patients. We have thus to conclude that PD-L1 alone is not the ideal biomarker and in current practice should not be systematically proposed to patients except in cisplatin-ineligible patients in a first-line setting. The major limitation of PD-L1 is the lack of standardization of the PD-L1 assay (see Table 1). Different immunohistochemistry assays are used for PD-L1 scoring across the trials, using different cell types (TCs, ICs or combinations of the two), distinct cut-off values for positivity (from 1% to 25%) and distinct antibodies. Even if a retrospective analysis of 235 UC samples showed a relative concordance in PD-L1 staining when using the different antibodies (22C3, 28-8, SP142, E1L3N), standardization in the PD-L1 staining may improve the design of further clinical trials and data collection in retrospective analyses. Furthermore, PD-L1 expression is dynamic during disease evolution, changing with time and therapies, and heterogeneity exists among primary tumors and metastases [42]. Most importantly, PD-L1 is only a single feature in the tumor microenvironment and could not reflect perfectly the antitumor capacity of the immune system.

Tumor mutational burden (TMB) is relatively high in UC compared with other cancers and reflects the production of neoantigens, which is, as described above, an important step in stimulating anti-tumoral immunity. TMB was also shown to be closely related to the immune microenvironment, suggesting that higher TMB tends to promote T-cell and NK infiltration into the tumor microenvironment [43]. In patients with NMIUC, a high TMB was significantly associated with a higher response rate and longer recurrence-free survival after BCG [44].

In cohort 2 of the IMvigor210 trial, TMB was higher in responding patients compared to non-responding patients (12.4 vs. 6.4 per megabase, respectively). In cohort 1 of this trial, high TMB was associated with better OS with atezolizumab compared to low TMB [20]. In a similar way, in the Checkmate 275 trial, patients with high TMB presented higher ORR (HR 2.13; 95% CI 1.26–3.60), PFS (HR 0.75; 95% CI 0.61–0.92) and OS (HR 0.73; 95% CI 0.58–0.91) compared to patients with low TMB [45]. The combination of TMB and PD-L1 was also found to have a better predictive value in term of PFS and OS compared to PD-L1 alone; in the IMvigor211 trial, median OS in patients with high TMB and high PD-L1 expression was higher with atezolizumab compared to chemotherapy (17.8 vs. 10.6 months) [46,47]. More challenging is the use of TMB in clinical practice due to the lack of technical standardization in terms of the cut-off determination and the panel of genes analyzed. The phase II NCT02553642 trial is currently ongoing to evaluate the relationship between TMB and response to nivolumab/ipilimumab in advanced UC.

Tumor gene expression can more accurately describe the immune tumor microenvironment by quantifying chemokines, cytokines, or cell surface proteins than PD-L1 and TMB. The interferon-gamma (IFN-γ) signature measures the expression of up to 25 genes involved in tumor-related inflammation. In the Checkmate 275 trial, the IFN-γ gene signature score was correlated with nivolumab efficacy; among the 59 patients with a high score, 20 presented CR/PR, compared to only 19 CR/PR among the 118 patients with a low/medium score [22]. Other gene signatures are currently evaluated in order to better identify clusters that could be associated with “hot” tumors, characterized by a high degree of immune infiltration, or “cold” tumors that could represent an immune desert. In the JAVELIN Bladder 100 study, expression of immune-related genes of the innate and adaptive immune system (CD8, IFN-γ, LAG3, TIGIT and CXCL9) and the number of alleles encoding high-affinity Fc gamma receptor variants predicted higher survival benefit with avelumab maintenance [28].

In the IMvigor130 trial, the apolipoprotein B editing catalytic polypeptide (APOBEC) signature was associated with a better response with atezolizumab treatment. The APOBEC enzymes are involved in DNA repair processes and are associated with mutation signature in UC. High APOBEC mutational signature was associated with a longer survival in atezolizumab arms (monotherapy or combination plus chemotherapy) compared to chemotherapy alone [48].

Of course, these gene signatures require validation in prospective trials. The Tumor Cancer Genome Atlas (TCGA) identified five classes of UC, depending on oncogenic mechanisms, infiltration by immune and stromal cells and histological features: luminal–papillary, luminal-infiltrated, luminal, basal–squamous and neuronal [49,50,51]. Recently, an international consensus molecular classification reconciled the different published classification schemes, including the TCGA. Basal–squamous, stroma-rich tumors and luminal non-specified tumors present an important immune infiltration and could thus be potential candidates for ICIs [52]. Large prospective trials will be required to evaluate the role of this classification in the prediction of ICI efficacy. The subgroup analysis of JAVELIN Bladder 100 showed that the benefit of maintenance with avelumab compared to BSC was not observed in luminal tumors but was apparent in basal–squamous (24 vs. 17.9 months), luminal infiltrated (19.9 vs. 14.3 months) and luminal–papillary tumors (22.5 vs. 13.4 months) [53].

## 6. Future Strategies with ICIs in Urothelial Carcinoma Based on Immune Resistance Mechanisms

Even if the expression of PD-L1 by tumor cells and APCs plays a key role in the development of immune evasion, many other mecanisms are involved. It was shown that bladder cancer was characterized by a highly immunosuppressive environment. Tumor cells secrete various immunosuppressive and anti-apoptotic factors (vascular endothelial growth factor (VEGF), TGF-β, IL-10 and IL-6, prostaglandine E2 (PGE2)) that create a tolerogenic microenvironment with accumulation of ICs harboring immunosuppressive phenotypes, such as myeloid-derived suppressor cells (MDSCs), tumor-associated macrophages (TAMs) and regulatory T-cells (Tregs). Figure 4 depicts different mechanisms of immune evasion. Acting on the tumor microenvironmenet could help to increase ICI efficacy (Table 2).

Among the immunosuppressive factors, PGE2 is a metabolite of the COX pathway that exhibits anti-apoptotic effects and stimulates the proliferation and renewal of bladder cancer stem cells. PGE2 also participates in immune evasion by inhibiting APC differentiation, stimulating MDSCs and recruiting Tregs to tumor sites. PGE2 was also shown to suppress IFN-γ and IL-2 production by T-cells and NK cells [54]. COX2 inhibitors could thus appear as potential candidates for ICI combinations.

Hyaluronan or hyaluronic acid (HA) is a cell surface glycoprotein that favors tumor progression by inducing tumor cell motility, invasive properties and proliferation. HA also regulates negatively the secretion of different cytokines and chemokines involved in cytotoxicity, participating in immune evasion [55,56].

MDSCs, DCs and tumor cells have the capacity, by autocrine IFN-g release, to secrete indoleamine 2,3-dioxygenase 1 (IDO1), an enzyme that catabolizes tryptophan in kynurenine. This metabolite impairs T-cell clonal expansion and activity. The combination of ICI and IDO inhibitors could thus improve the efficacy of ICIs and delay resistance to these agents. Linrodostat mesylate, an oral IDO1 inhibitor, has been evaluated in combination with nivolumab in mUC in 27 ICI-naïve patients in a phase I/II trial (NCT02658890); the ORR reached 37% with a disease control rate of 56% [57]. Larger randomized trials are needed.

The combination of ICIs plus agents targeting oncogenic mutations in mUC could also be promising. PI3K belongs to the PI3K–Akt–mTOR cascade and plays a role in maintaining the immunosuppressive role of macrophages and MDSCs. The randomized phase II MARIO-275 (NCT03980041) trial evaluated the safety and efficacy of the combo of the PI3K-inhibitor eganelisib plus nivolumab in ICI-naïve mUC patients. The ORR was higher in the eganelisib plus nivolumab combo arm compared to the placebo plus nivolumab arm in patients with low MDSC levels (38.5% vs. 23.1%, respectively) and in PD-L1-negative patients (26.1% vs. 14.3%, respectively) [58]. Around 15% of mUC patients harbor alterations in fibroblast growth factor receptor (FGFR), resulting in excessive stimulation of the downstream signaling cascade [59]; FGFR3 was also associated with a T-cell-depleted tumor environement and FGFR inhibition with erdafitinib was demonstrated to upregulate inflammation-related signaling. Different trials are curently evaluating the efficacy of FGFR inhibitors plus ICI combos in mUC.

Oncogenic activation of angiogenesis, resulting in abnormal vasculature, enhances immune escape of cancer cells. Excessive stimulation of pro-angiogenic factors, such as VEGF, mainly affects migration and activation of ICs [60]. Antiangiogenic agents could, on the one hand, abolish the nutrient and oxygen supply to the TCs but, on the other hand, potentialize ICI [61]. Different phase I/II trials are currently ongoing in order to evaluate the efficacy of antiangiogenic agents plus ICI combinations in different cancer types. For example, in the NCT02496208 phase I trial, cabozantinib, a tyrosine kinase inhibitor targeting mainly VEGFR and c-MET, is being evaluated in combination with nivolumab +/− ipilimumab in different genitourinary cancers, including mUC.

The combination of ICI plus radiotherapy (RT) is currently being evaluated in clinical trials. RT can potentialize the anticancer effect of ICIs by stimulating immunogenic cancer cell death. RT-induced release of neoantigens stimulates APCs and subsequent T-cell activation. Preclinical trials in different cancer cell types have shown that RT in combination with ICIs could result in high anticancer efficacy by enhancing TILs-mediated anti-tumor immunity and by stimulating macrophage repolarization with increases in M1/M2 ratios [62]. Combining RT with ICI could also stimulate the abscopal effect of RT, a phenomenon by which systemic anti-tumor responses are observed outside of the primary site of local irradiation. In a randomized phase I trial, pembrolizumab in combination with concomittant stereotactic body radiotherapy (SBRT) to the largest metastatic lesion in mUC patients resulted in higher response and better OS compared to pembrolizumab given sequentially with SBRT (44% vs. 0% and 12.1 vs. 4.5 months, respectively) [63]. Different trials are currently ongoing to determine the optimal doses of RT, the optimal ICI agent as well as the optimal timing of treatment.

Up to 40% of UC patients harbor mutations in at least one gene of the DNA-damage repair (DDR) system, including *ERCC2, BRCA1, ATM, CHEK2, BRCA2* and *PALB2*. Alterations in DDR genes are associated with high mutation load, increased TILs and higher response to ICIs [64]. Despite low anticancer efficacy in monotherapy, PARP inhibitors could have the potential to enhance the activity of ICIs in mUC in some patients [65]. The phase II BAYOU trial evaluated the addition of olaparib to durvalumab in patients with previously untreated, platinum-ineligible mUC. Patients were randomized to receive either durvalumab plus olaparib or durvalumab alone. In patients with homologous recombination repair (HRR) mutations, the combo (*n* = 17) reached higher PFS compared to durvalumab alone (*n* = 14) (5.6 vs. 1.8 months, respectively; HR 0.18; *p* < 0.001) [66]. The TALASUR trial (NCT04678362) is currently evaluating the benefit of associating a PARP inhibitor with avelumab in a maintenance setting.

Antibody–drug conjugates (ADCs) are composed of monoclonal antibodies linked to cytotoxic drugs; this allows a specific delivery to cells that express the target antigen of the selected antibody [67,68]. In the randomized phase III EV-301 trial, enfortumab vedotin (EV) was compared to chemotherapy (paclitaxel, docetaxel or vinflunine) in mUC patients after failure of platinum-based therapy and ICI. EV resulted in longer median OS compared to chemotherapy (12.88 vs. 8.97 months; HR 0.70; 95% CI 0.56 to 0.89; *p* = 0.001) [69]. The randomized phase III study is currently evaluating the benefit of combining EV and pembrolizumab in previously untreated mUC patients (EV-302, NCT04223856).

## 7. Conclusions

The place of ICIs is continuously evolving, having initially demonstrated efficacy after the failure of platinum-based therapy and representing now the standard in maintenance settings. This is a new concept in mUC: introducing ICIs directly after platinum-based therapy, without waiting for disease progression, increases the survival of patients compared to historical sequential treatment. ICIs appeared as the optimal agent for this approach to maintenance, as tolerance is much better compared to chemotherapy or targeting agents. Longer follow-ups and phase IV trials with evaluation of worldwide experience will help to confirm the role of the maintenance strategy in the management of mUC. Further trials will also confirm the place of new anticancer agents, such as antibody–drug conjugates, in monotherapy or in combination with ICIs in the treatment of this aggressive cancer.

## Figures and Tables

**Figure 1 cancers-14-01640-f001:**
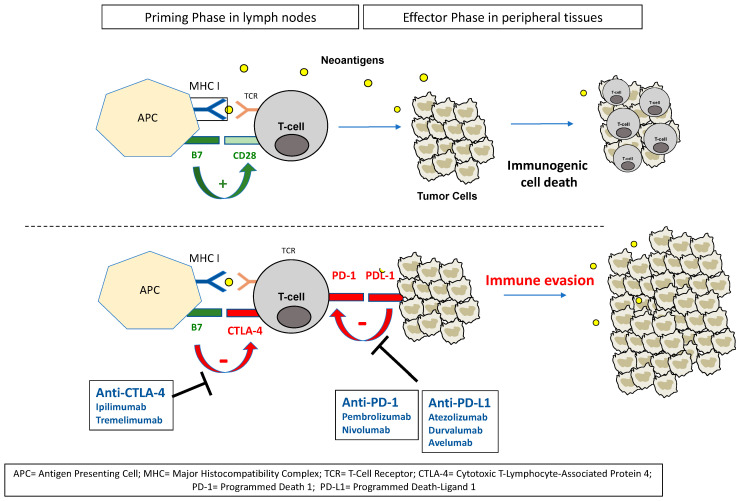
The priming phase consists of the activation of T-cells by antigen-presenting cells (APCs) in lymph nodes. These neoantigens allow the binding of MHCs on APCs to TCRs on T-cells, inducing co-activation signals (B7-CD28) that result in the activation of T-cells. Activated T-cells, in turn, infiltrate tumors and kill tumor cells by enhancing inflammatory reactions. Malignant cells develop different mechanisms to evade immune recognition, including the upregulation of immune checkpoints, such as CTLA-4 and PD-1, on tumor-specific lymphocytes, and PD-L1 on tumor cells themselves. The binding of these immune chekpoints leads to decreased activity of immunological action of T-cells and impairs their capacity to infiltrate tumors and activate inflammatory reactions.

**Figure 2 cancers-14-01640-f002:**
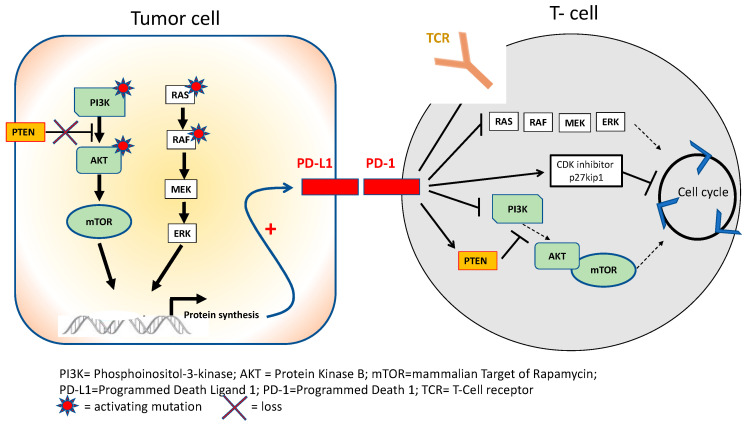
In tumor cells, oncogenic activation of PI3K–Akt–mTOR and MAPK and loss of PTEN induces overexpression of PD-L1 via stimulation of protein synthesis. In T-cells, PD-1 expression inhibits downstream activation of the PI3K–Akt–mTOR and MAPK pathways, inhibits cell cycle progression and negatively regulates T-cell receptor activity.

**Figure 3 cancers-14-01640-f003:**
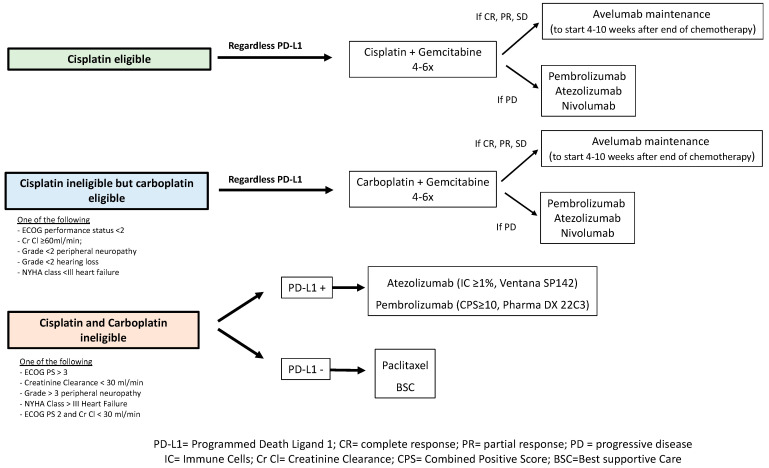
Current management of metastatic UC in a first-line setting.

**Figure 4 cancers-14-01640-f004:**
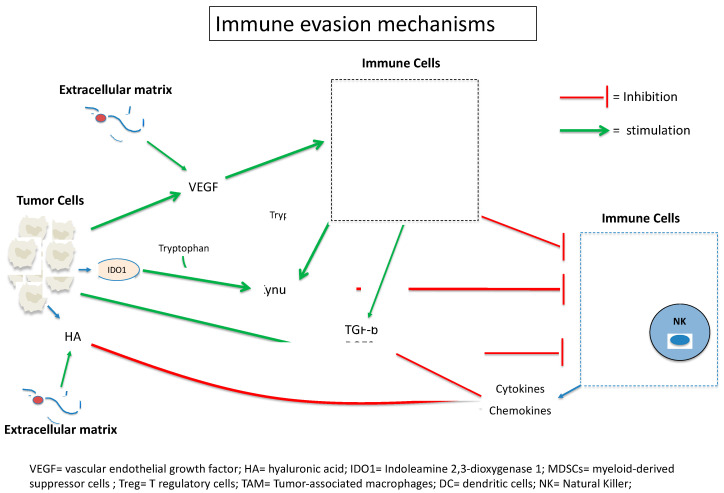
Bladder cancer was characterized by a highly immunosuppressive environment. Tumor cells secrete various immunosuppressive and anti-apoptotic factors (vascular endothelial growth factor (VEGF), TGF-β, IL-10 and IL-6, prostaglandine E2 (PGE2)) that create a tolerogenic microenvironment with accumulation of ICs harboring immunosuppressive phenotypes.

**Table 1 cancers-14-01640-t001:** Pivotal trials that evaluated immune checkpoint inhibitors in advanced urothelial carcinoma.

Trial	Setting	Agent	*N*	ORR	OS (Months)	PD-L1 Positivity
Phase II Single-agent IMvigor210 Cohort 2 [20]	After failure of platinum-based therapy	Atezolizumab	310	All pts = 16%, CR = 7% IC2/3 PD-L1 = 28%, CR = 15%	All pts = 7.9; 1 y OS = 37% High PD-L1 = 11.9; 1 y OS = 50%	PD-L1 on tumor-infiltrating ICs: IC0 (<1%), IC1 (≥1% but <5%), and IC2/3 (≥5%) (Ventana SP142)
Phase III Randomized IMvigor211 [21]	After failure of platinum-based therapy	Atezolizumab	467	All pts = 14%, CR = 4% High PD-L1 = 23%	All pts = 8.6 (*p =* 0.038); 1 y OS = 40% High PD-L1 = 11.1 (*p* = 0.41)	PD-L1 on tumor-infiltrating ICs: IC0 (<1%), IC1 (≥1% but <5%), and IC2/3 (≥5%) (Ventana SP142)
		Vinflunine or paclitaxel or docetaxel	464	All pts = 15%, CR = 4% High PD-L1 = 22%	All pts = 8.0; 1 y OS = 33% High PD-L1 = 10.6	
Phase II Single-agent Checkmate 275 [22]	After failure of platinum-based therapy	Nivolumab	265	All pts = 19.6% High PD-L1 = 28.4% Low PD-L1 = 16.1%	All pts = 8.7; 1 y OS = 41% High PD-L1 = 11.3; Low PD-L1 = 5.9	PD-L1 on TCs: high (≥5%), medium (≥1%) and low (<1%) (Pharma Dx 28-8)
Phase III Randomized KEYNOTE-045 [23,24]	After failure of platinum-based therapy	Pembrolizumab	270	All pts = 21.1%, CR = 7%	All pts = 10.1 High PD-L1 = 8	CPS = percentage of PD-L1 in ICs and TCs related to numbers of TCs. Positive if ≥10 (Pharma Dx 22C3)
		Vinflunine or Paclitaxel or Docetaxel	272	All pts = 11.4%, CR = 3.3%	All pts = 7.4 High PD-L1 = 5.2	
Phase I/II Single agent [25,26]	After failure of platinum-based therapy	Durvalumab	182	All pts = 17% High PD-L1 = 26.3% Low PD-L1 = 4.1%	All pts = 14.1; 1 y OS = 50%	PD-L1 positivity if ≥25% of TCs or ≥25% of ICs if >1% of the tumor area contained ICs or 100% of ICs if ≤1% of the tumor area contained ICs (Ventana SP263)
Phase Ib Single agent (Javelin) [27]	After failure of platinum-based therapy	Avelumab	242	All pts = 16.1% High PD-L1 = 25% Low PD-L1 = 14.7%	All patients = 7.4; 1 y OS = 54.9%	PD-L1 on TCs: Positive if ≥5% (Ventana SP263)
Phase III Randomized JAVELIN Bladder 100 [28]	Maintenance setting after response or stable disease on first-line platinum-based therapy	Avelumab	350	All pts = 9.7%, CR = 6% High PD-L1 = 13.8%, CR = 9.5%	All pts = 21.4 (*p* = 0.0005); 1 y OS = 71.3% High PD-L1 = NR; 1 y OS = 79.1%	PD-<L1 positivity if ≥25% of TCs or ≥25% of ICs if >1% of the tumor area contained ICs or 100% of ICs if ≤1% of the tumor area contained ICs (Ventana SP263)
		BSC	350	All pts = 1.4%, CR = 0.9% High PD-L1 = 1.2%, CR = 0.6%	All pts = 14.3; 1 y OS = 58.5% High PD-L1 = 17.1; 1 y OS = 60.4%	
Phase III Randomized IMvigor130 [29]	1st-line mUC	Atezolizumab + plt/gem	451	All pts = 48%, CR = 13%	All pts = 16 (*p* = 0.027 but did not cross boundary *p*-value 0.007)	PD-L1 on tumor-infiltrating ICs: IC0 (<1%), IC1 (≥1% but <5%), and IC2/3 (≥5%) (Ventana SP142)
		Atezolizumab	362	All pts = 23%, CR 6%	All pts = 15.7 High PD-L1 = 27.5	
		Placebo + plt/gem	400	All pts = 44%, CR = 7%	All pts = 13.4	
Phase III Randomized KEYNOTE-361 [30]	1st-line mUC	Pembrolizumab + plt/gem	351	All pts = 54.7%, CR = 15%, High PD-L1 = 57.2%, CR = 16%	All pts = 17 (*p* = 0.0407 but did not cross boundary p-value 0.014 1 y OS = 62%	CPS = percentage of PD-L1 in ICs and TCs related to numbers of TCs. Positive if ≥10 (Pharma Dx 22C3)
		Pembrolizumab	307	All pts = 30.3%, CR = 11% High PD-L1 = 32.5%, CR = 13%	All pts = 15.6 High PD-L1 = 16.1	
		Plt/gem	352	All pts = 44.9%, CR = 12% High PD-L1 = 46.2%, CR = 17%	All pts = 14.3 1 y OS = 56%	
Phase III Randomized DANUBE [31]	1st-line mUC	Durvalumab + Tremelimumab	342	All pts = 36%, CR = 8% High PD-L1 = 47%, CR = 12%	All pts = 15.1 (*p* = 0.075) High PD-L1 = 17.9	PD-<L1 positivity if ≥25% of TCs Or ≥25% of ICs if >1% of the tumor area contained ICs Or 100% of ICs if ≤1% of the tumor area contained ICs (Ventana SP263)
		Durvalumab	346	All pts = 26%, CR = 8% High PD-L1 = 28%, CR = 10%	All pts = 13.2 High PD-L1 = 14.4	
		plt/gem	344	All pts = 49%, CR = 6% High PD-L1 = 48%, CR = 7%,	All pts = 12.1 High PD-L1 = 12.1	
Phase II Single-agent IMvigor210 Cohort 1 [20,32,33]	1st-line mUC (cisplatin ineligible)	Atezolizumab	119	All pts = 24%, CR = 7% High PD-L1 = 24%	All pts = 15.9 High PD-L1 = 12.3 Low PD-L1 = 19.1	PD-L1 on tumor-infiltrating ICs: IC0 (<1%), IC1 (≥1% but <5%), and IC2/3 (≥5%) (Ventana SP142)
Phase II Single-agent KEYNOTE-052 [34,35]	1st-line mUC (cisplatin ineligible)	Pembrolizumab	370	All pts = 29%, CR = 7%	All pts = 11.3 High PD-L1 = 18.5	CPS = percentage of PD-L1 in ICs and TCs related to numbers of TCs. Positive if ≥10 (Pharma Dx 22C3)
Phase I–II CheckMate 032 [36]	After failure of platinum-based therapy	N 1 mg/kg + I 3 mg/kg	61	All pts = 23%, CR = 2%	All pts = 10.2	PD-L1 on TCs: high (≥5%), medium (≥1%) and low (<1%) (Pharma Dx 28-8)
		N 3 mg/kg + I 1 mg/kg	54	All pts = 19%, CR = 0%	All pts = 7.3	

Pts = patients; PD-L1 = programmed death ligand 1; ICs = immune cells; OS = overall survival; TCs = tumor cells; 1 y = 1 year; CR = complete response; CPS = combined positive score; mUC = metastatic urothelial carcinoma; N = nivolumab; I = ipilimumab.

**Table 2 cancers-14-01640-t002:** Overview of important trials evaluating potential combinations with immune checkpoint inhibitors in metastatic urothelial carcinoma.

Trial	Cancer	Associated Agent	ICI
Phase I/II (NCT02658890) [54]	Second- or later line mUC	Inhibitor of IDO (Linrodostat mesylate)	Nivolumab
Phase II NCT03915405	Second- or later line mUC	Inhibitor of IDO (KHK2455)	Avelumab
Phase II MARIO-275 (NCT03980041) [55]	Second- or later line mUC ICI-naïve	PI3K inhibitor Eganelisib	Nivolumab
Phase Ib/II trial (NCT03473743)	First-line treatment in mUC (cisplatin ineligible)	FGFR inhibitor Erdafitinib	Cetrelimab (anti-PD-1)
Phase Ib/II trial FORT-2 (NCT03473756)	First-line treatment in mUC (cisplatin ineligible)	FGFR inhibitor Rogaratinib	Atezolizumab
Phase Ib/II trial FIERCE-22 (NCT03123055)	Second- or later line mUC	FGFR3 inhibitor Vofatamab	Pembrolizumab
Phase I NCT02496208	Second- or later line Advanced Genito-urinary cancers	Antiangiogenic tyrosine kinase inhibitor Cabozantinib	Nivolumab +/− Ipilimumab
Phase I/II NCT03170960	First- or later line Advanced metastatic cancer	Antiangiogenic tyrosine kinase inhibitor Cabozantinib	Atezolizumab
Phase II (NCT03601455)	Locally advanced or metastatic UC; Ineligible for chemotherapy or refusing chemotherapy	External Beam RT (5 fractions beginning on day 8 of cycle 1 of ICI)	Durvalumab +RT vs. Durvalumab + tremelimumab + RT
Phase II (NCT03115801)	Previously treated mUC with ≥2 metastatic sites ICI-naïve patient	RT on 1 lesion (30 Gy in 3 fractions of 10 Gy)	ICI (nivolumab, atezolizumab, pembrolizumab) vs. ICI + RT
Phase II (NCT03693014)	Previously treated metastatic cancer of any histology with limited progression on ICI	SBRT	Continuing ICI
Phase II NCT03486197	mUC ≥2 metastatic sites	Neutron-based RT 3 × 5 fractions over 2 weeks	Pembrolizumab
Phase II TALASUR (NCT04678362)	mUC in maintenance setting	Talazoparib	Avelumab in maintenance in platinum-sensitive mUC patients
Randomized Phase III NCT04223856	Previously untreated mUC	Enfortumab Vedotin	Pembrolizumab

Legend: mUC = metastatic urothelial carcinoma; IDO1 = indoleamine 2,3-dioxygenase 1; RT = radiation therapy; SBRT = stereotaxic radiation therapy; ICI = immune checkpoint inhibition; PI3K = phosphoinositol 3-kinase; FGFR = fibroblast growth factor receptor.

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
