# Peer review of "The Evolution of Immune Checkpoint Inhibitors in Advanced Urothelial Carcinoma"

_cancers, 2022, doi:10.3390/cancers14071640_

Round 1

Reviewer 1 Report

This review aimed to assess the current ICI managements of mUC of the bladder and give readers a summary of ICI therapies for mUC according to the clinical trials. The manuscript is generally well written; however, some critical points mentioned below should be noticed and revised before publication.

*In the abstract, P.1, line 10-12: “If these agents were first used in monotherapy after failure of platinum-based chemotherapy, new strategies have been evaluated, including association of ICIs, ICI plus chemotherapy association or maintenance therapy.” I don’t quite understand this sentence. Do the authors mean that new strategies have replaced ICI monotherapy as preferred therapies after failure of platinum-based chemotherapy?

*In most of sections, the authors only presented the data of clinical trials and lack of the interpretation and summary, which will be helpful for the readers. For example, P.11, line 308, section “ICI monotherapy should not be proposed in first-line setting in cisplatin-ineligible patients”: The authors offered a lot of clinical data from the ICIs trials and state that use of carboplatin-based chemotherapy followed by avelumab maintenance is as good as using ciplatin-based chemotherapy followed by avelumab. However, they did not interpret the data and tell the readers why ICI monotherapy should not be proposed in first-line setting in cisplatin-ineligible patients. I would suggest the authors give a summary at the end of each section.

*P.7, line 88: “molecular rational­ for clinical efficacy” What does “rational” mean?

*What does “If PD” in Figure 3 mean?

*P.8, line 141-142: Despite the absence of superiority of atezolizumab compared to chemotherapy and the absence of level 1, these results suggest…” Do you mean level 1 evidence?

*The abbreviation should be defined upon first use (e.g. TMB)

*English writing should be reviewed by a native speaker.

Author Response

*In the abstract, P.1, line 10-12: “If these agents were first used in monotherapy after failure of platinum-based chemotherapy, new strategies have been evaluated, including association of ICIs, ICI plus chemotherapy association or maintenance therapy.” I don’t quite understand this sentence. Do the authors mean that new strategies have replaced ICI monotherapy as preferred therapies after failure of platinum-based chemotherapy?

You right : This sentence was not clear. I completely change change by "These agents were first used in monotherapy after failure of platinum-based chemotherapy, but different strategies have been evaluated exploring the use of ICIs in first-line metastatic setting. The “maintenance” strategy consists in the introduction of ICI in patients who experienced benefit from first-line chemotherapy in metastatic setting."

*In most of sections, the authors only presented the data of clinical trials and lack of the interpretation and summary, which will be helpful for the readers. For example, P.11, line 308, section “ICI monotherapy should not be proposed in first-line setting in cisplatin-ineligible patients”: The authors offered a lot of clinical data from the ICIs trials and state that use of carboplatin-based chemotherapy followed by avelumab maintenance is as good as using ciplatin-based chemotherapy followed by avelumab. However, they did not interpret the data and tell the readers why ICI monotherapy should not be proposed in first-line setting in cisplatin-ineligible patients. I would suggest the authors give a summary at the end of each section.

short summarizes were added for each part of the manuscript

For the maintenance strategy

The fact that carboplatin does not alter subsequent efficacy of avelumab maintenance and that 4 cycles of chemotherapy were found to be similar to 6 cycles is of benefit for patients who are deemed ineligible for cisplatin and/or who tolerate poorly chemotherapy. Early introduction of avelumab allows for all patients to receive both chemotherapy and ICI; this is particularly interesting in patients with high tumor burden who experienced only stable disease with chemotherapy.

Superiority of ICI over platinum-based chemotherapy has not been demonstrated in first-line metastatic setting

These three trials did not show any superiority of ICI monotherapy compared to platinum-based chemotherapy in patients with mUC in first line setting. ICI should not be considered in patients deemed eligible for platinum-based chemotherapy, particularly in case of high tumor burden if rapid important response is needed (Figure 3). Once again, the results of JAVELIN bladder 100 trial favor the introduction of platinum-based chemotherapy (cisplatin or carboplatin) in first-line setting followed by avelumab maintenance.

No evidence of superiority of ICI monotherapy over carboplatin in first-line setting in cisplatin-ineligible patients

Head-to-head comparison with carboplatin-based regimen is not feasible; however, the higher response rate observed with carboplatin suggests to prefer carboplatin regimen over ICI in cisplatin-ineligible patients with high tumor burden. This strategy is also support by the JAVELIN Bladder 100 results. Benefit was improved both with cisplatin and with carboplatin if avelumab maintenance is further considered; the HR for patients receiving avelumab maintenance compared to BSC after cisplatin- and after carboplatin-based regimen was 0.69 (0.51-0.94) and 0.66 (0.47-0.91), respectively

A minority of mUC patients are deemed ineligible for carboplatin in frontline; carboplatin ineligibility involve patients with poor general status (ECOG PS > 3), creatinine clearance < 30 ml/min), grade > 3 peripheral neuropathy and severe heart failure. In these patients, as options are very limited, ICI could be proposed in first-line setting, with higher benefit expected in PD-L1 positive patients (Figure 3).

*What does “If PD” in Figure 3 mean?

 PD = progressive disease; added

*P.7, line 88: “molecular rational­ for clinical efficacy” What does “rational” mean?

right : no place for molecular

*P.8, line 141-142: Despite the absence of superiority of atezolizumab compared to chemotherapy and the absence of level 1, these results suggest…” Do you mean level 1 evidence?                                             

yes= level 1 evidence

The abbreviation should be defined upon first use (e.g. TMB)

done

Reviewer 2 Report

The authors present an extensive and exquisite review about checkpoint inhibitors in urothelial cancer. The text is well written, easy to follow and properly divided. Some minor language issues might be upgraded by a native speaker. In my opinion, the paper may be accepted in present form.

Author Response

thank you for your comments

We corrected minor changes

Reviewer 3 Report

This is a nice comprehensive review paper on ICI for advanced UC.    The authors may want to address the following points before publication. 1_ The authors comprehensively and thoroughly describe current practice and clinical trials on the usage of ICIs for advanced UC patients in the main text. However, the reviewer wonders if it is unbalanced that a maintenance ICI treatment with avelumab is too emphasized in the conclusion compared with actual clinical impact. The authors may want to modify their conclusions to tone down the impact of a maintenance ICIs. 2_ Table 1: The authors may want to provide p values for prospective controlled studies for the readers to easily understand the significance of important clinical trials. 3_ Lines 239-41: Although the authors mention “no statistical difference”, a p value of the outcomes described is <0.05. Please make sure if the description is correct. 4_ Line 297: “lower” may be a typographic error. Please check it. 5_ The authors use a term “association” when they describe combination of two or more systemic treatment modalities. The authors may want to use “combination” or “combo” for better understanding by the readers.

Author Response

1_ The authors comprehensively and thoroughly describe current practice and clinical trials on the usage of ICIs for advanced UC patients in the main text. However, the reviewer wonders if it is unbalanced that a maintenance ICI treatment with avelumab is too emphasized in the conclusion compared with actual clinical impact. The authors may want to modify their conclusions to tone down the impact of a maintenance ICIs.

We tone down the role of maintenance with avelumab, with however, highlighting the benefit of this strategy in mUC.

“Longer follow-up and phase IV trial with evaluation of worldwide experience will help to confirm the role of maintenance strategy in the management of mUC. Further trials will also confirm the place of new anticancer agents such as antibody-drug conjugate in this aggressive cancer.”

2_ Table 1: The authors may want to provide p values for prospective controlled studies for the readers to easily understand the significance of important clinical trials.

done

3_ Lines 239-41: Although the authors mention “no statistical difference”, a p value of the outcomes described is <0.05. Please make sure if the description is correct.

The criteria for significant p-value per protocol was added

“In the ITT population, PFS tended to be increased in group A compared to group C (8.2 vs 6.3 months, respectively; P=0.007). There was no statistical difference in median OS between group A and C, with a median survival of 16 months and 13.4 months respectively (HR 0.83, 95% CI 0.69-1.00; P=0.027 but did not cross the interim efficacy boundary of 0.007)”

4_ Line 297: “lower” may be a typographic error. Please check it.

Right. I replace by: “There was no significant difference in the median OS between durvalumab and chemotherapy in the PD-L1-positive population (14.4 vs 12.1 months, respectively; HR: 0.89, 95% CI 0.71-1.11; P=0.30).”

5_ The authors use a term “association” when they describe combination of two or more systemic treatment modalities. The authors may want to use “combination” or “combo” for better understanding by the readers.

we understand this comment and change in “combination”

Reviewer 4 Report

Manuscript entitled "The evolution of immune checkpoint inhibitors in advanced urothelial carcinoma"

This work is merit and could be acceptable pending some modification. The authors are encouraged to:

  1. Describe more about the mechanisms for immune evasion. Probably an illustration should be made.
  2. It is getting clear that ICI could be better to be used in an combinaiton therapy manner. The authors should discuss more about this issue and profile the biological significance.

In the current work, when clinical trials are mentioned, the authors are very sticky to PD-1 and PD-L1 therapeutics with/without chemotherapy and one with CTLA-4 targeted therapy.

While the authors are encouraged to include those studies with combined radiation therapy. Moreover, many ongoing trials in which PD-1/PD-L1 Ab combined with PARPi, ADC, HDACi, anti-KIRs, ... ... had been proposed.

The authors must add these as separate tables to make this work completed. 

Post hoc analysis for various predictive biomarkers should also be mentioned.

Author Response

please see the attachement

Round 2

Reviewer 1 Report

I would suggest the authors reconsider the adequacy of some subtitles, such as:

“4.4 No evidence of superiority of ICI monotherapy over carboplatin in first-line setting in cisplatin-ineligible patients”

“4.5 No role for ICIs combination in mUC today, without any predictive biomarker”

“6. Immune evasion, resistance to ICI and future combinations in urothelial carcinoma”

These subtitles are confusing and may mislead the readers.

Author Response

4.4 No evidence of superiority of ICI monotherapy over carboplatin in first-line setting in cisplatin-ineligible patients”

==> change in "The place of ICI in cisplatin-ineligible patients"

4.5 No role for ICIs combination in mUC today, without any predictive biomarker”

==> change in "The place of ICI-ICI combination in mUC ?"

6. Immune evasion, resistance to ICI and future combinations in urothelial carcinoma”

==> change in "Future strategies with ICI in urothelial carcinoma, based on immune resistance mechanisms"

Reviewer 4 Report

I have no further comment. This work is acceptable in the present form.

Author Response

thank you

Round 3

Reviewer 1 Report

I don’t think the current changed 4.4 and 4.5 subtitles would be better than the previous ones. The subtitles should be clear-cut and tell the readers what the main point of the section is.

Author Response

4.4 Strategies in cisplatin-ineligible patients: monotherapy or maintenance ICI  

4.5 No current place fo ICI-ICI combination in mUC